# Down-regulated GAS6 impairs synovial macrophage efferocytosis and promotes obesity-associated osteoarthritis

Zihao Yao[1,2†], Weizhong Qi[1,2†], Hongbo Zhang[1,2†], Zhicheng Zhang[1,2], Liangliang Liu[1,2], Yan Shao[1,2], Hua Zeng[1,2], Jianbin Yin[1,2], Haoyan Pan[1,2], Xiongtian Guo[1,2], Anling Liu[3], Daozhang Cai[1,2*], Xiaochun Bai[1,2*], Haiyan Zhang[1,2*]

[1]Department of Orthopedics, Academy of Orthopedics·Guangdong Province, Guangdong Provincial Key Laboratory of Bone and Joint Degeneration Diseases, The Third Affiliated Hospital of Southern Medical University, Guangzhou, China; [2]Department of Joint Surgery, Center for Orthopedic Surgery, Orthopedic Hospital of Guangdong Province, The Third School of Clinical Medicine, Southern Medical University, The Third Affiliated Hospital of Southern Medical University, Guangzhou, China; [3]Department of Biochemistry and Molecular Biology, School of Basic Medical Sciences, Southern Medical University, Guangzhou, China

**\*For correspondence:**
cdz@smu.edu.cn (DC);
baixc15@smu.edu.cn (XB);
zhhy0704@126.com (HZ)

[†]These authors contributed equally to this work

**Abstract** Obesity has always been considered a significant risk factor in osteoarthritis (OA) progression, but the underlying mechanism of obesity-related inflammation in OA synovitis remains unclear. The present study found that synovial macrophages infiltrated and polarized in the obesity microenvironment and identified the essential role of M1 macrophages in impaired macrophage efferocytosis using pathology analysis of obesity-associated OA. The present study revealed that obese OA patients and $Apoe^{-/-}$ mice showed a more pronounced synovitis and enhanced macrophage infiltration in synovial tissue, accompanied by dominant M1 macrophage polarization. Obese OA mice had a more severe cartilage destruction and increased levels of synovial apoptotic cells (ACs) than OA mice in the control group. Enhanced M1-polarized macrophages in obese synovium decreased growth arrest-specific 6 (GAS6) secretion, resulting in impaired macrophage efferocytosis in synovial ACs. Intracellular contents released by accumulated ACs further triggered an immune response and lead to a release of inflammatory factors, such as TNF-α, IL-1β, and IL-6, which induce chondrocyte homeostasis dysfunction in obese OA patients. Intra-articular injection of GAS6 restored the phagocytic capacity of macrophages, reduced the accumulation of local ACs, and decreased the levels of TUNEL and Caspase-3 positive cells, preserving cartilage thickness and preventing the progression of obesity-associated OA. Therefore, targeting macrophage-associated efferocytosis or intra-articular injection of GAS6 is a potential therapeutic strategy for obesity-associated OA.

## Editor's evaluation

In this study, the authors demonstrated that patients with obese-OA and mice with ApoE deficiency showed phenotypes of synovitis and enhanced macrophage infiltration in synovial tissues. GAS6 secretion is decreased during M1 macrophage polarization during obese-OA, leading to impaired macrophage efferocytosis in synovial apoptotic cells. Intra-articular injection of GAS6 restored the phagocytic capacity of macrophages, decreased synovial cell apoptosis, and prevented OA progression in obese-OA mice.

## Introduction

Osteoarthritis (OA) is a common, chronic, degenerative joint disease and a significant cause of joint pain and even disability (*Hunter and Bierma-Zeinstra, 2019*). Epidemiological investigations have documented that obesity is one of the significant risk factors for OA (*Martel-Pelletier et al., 2016*; *Teichtahl et al., 2015*). A meta-analysis of joint replacements in obese patients in 2010 has shown that the risk of knee OA was five times higher in obese patients than in healthy individuals (*Clement and Deehan, 2020*). At the same time, being 'overweight' (i.e., obesity) doubled the proportion of joint replacement treatments required later in life (*Franklin et al., 2009*). At present, the incidence of obesity continues to increase (*Caballero, 2019*). It is estimated that by 2025, the global incidence of obesity will reach 18% in men and 21% in women (*NCD Risk Factor Collaboration, 2016*). Thus, elucidating the mechanisms by which obesity promotes OA development is essential for OA prevention and treatment.

It was initially believed that obesity affects OA by changing certain mechanical factors. However, the progression of OA continues in non-weight-bearing areas, even after the line of the force is corrected (*Yusuf et al., 2010*). Moreover, these obesity-related mechanical factors cannot be justified in the development of OA in non-weight-bearing joints such as the hands (*Pottie et al., 2006*). Scientists have paid specific attention to the effects and roles of various pro-inflammatory cytokines and adipokines in obesity (*Neumann et al., 2016*). Clinical data have confirmed that obese OA patients are often associated with severe chronic synovitis, which plays an essential role in the pathogenesis and progression of OA (*Sellam and Berenbaum, 2010*; *Mathiessen and Conaghan, 2017*). Our previous findings have indicated that synovial macrophage polarization is significantly correlated with synovitis in OA progression (*Koski et al., 2006*; *Daghestani et al., 2015*; *Zhang et al., 2018*). When synovitis occurs, macrophages are stimulated by various cytokines and consequently release inflammatory mediators (*Sun et al., 2020*). At the same time, excess energy caused by obesity leads to changes in various cell functions, including angiogenesis and inflammatory cell infiltration (*Collins et al., 2021*). However, the effect of obesity on synovial hyperplasia and macrophage polarization in OA development has not been reported yet. We speculate that changes in tissues and organs caused by obesity may also affect synovitis, which in turn affects the process of OA.

GAS6 is a secreted glycoprotein widely expressed throughout the body. It is well known for its vital role in bridging phosphatidylserine on the surface of apoptotic cells (ACs) with its receptors Tyro3, Axl, and Mer, triggering the engulfment of ACs in an inflammatory environment (*Bellan et al., 2019*). This macrophage-related phagocytic process is also known as 'efferocytosis', which is beneficial for resolving inflammation (*Doran et al., 2020*). Prior studies have shown that impaired efferocytosis weakens the ability to clear ACs, inducing the release of inflammatory factors and ultimately causing synovitis (*Nepal et al., 2019*). Nevertheless, obesity-related macrophage polarization and the effect of obesity on efferocytosis remain unclear.

The present study found that obese OA patients and obese $Apoe^{-/-}$ mice are more prone to M1 macrophage infiltration in synovial tissue. Obese OA mice had more severe cartilage destruction and increased synovial ACs than OA mice in the control group. Down-regulation of GAS6 by M1 macrophages resulted in impaired efferocytosis for synovial ACs, causing synovial hyperplasia and obesity-associated OA development. These data suggested that targeting macrophage phagocytosis and polarization in obese patients with OA may be a potential therapeutic strategy.

**Table 1.** Blood lipids in obesity-related patients and general patients.

|  | Normal individuals | OA patients without obesity | Obese individuals | OA patients with obesity |
|---|---|---|---|---|
| Total cholesterol (TC) (3.0–6.0) | 4.81 ± 0.7 | 4.7 ± 0.54 | 6.34 ± 0.79 | 6.2 ± 0.86 |
| Triglyceride (TG) (0.56–1.7) | 1.34 ± 0.32 | 1.45 ± 0.21 | 4.3 ± 0.25 | 3.19 ± 0.48 |
| BMI index | 22.9 ± 0.35 | 23.5 ± 0.22 | 28.6 ± 0.17 | 28.4 ± 0.21 |

## Results

### Synovial tissues are highly hyperplastic in obese OA patients and infiltrated with more polarized M1 macrophages than non-obese OA patients

To investigate the role of obesity in synovial tissue in OA patients, levels of total cholesterol, triglycerides, and body mass index were examined in different patients. All subjects were divided into the following four groups based on the obtained values for these three factors: (1) normal individuals, (2) OA patients without obesity, (3) obese individuals, and (4) OA patients with obesity (*Table 1*). Consistent with our previous study, highly hyperplastic synovial tissues and abundant inflammatory cell infiltration were observed in human OA synovial tissue, combined with a significantly higher synovitis score than normal controls. Interestingly, the synovium tended to be hyperplastic in obese patients and reached a maximum in obese OA patients among the four groups (*Figure 1A, C*).

We further investigated the polarization level of macrophages in synovial tissues by staining with F4/80 (macrophage markers), inducible nitric oxide synthase (iNOS; M1 macrophage marker), and CD206 (M2 macrophage marker). As a result, the number of M1 macrophages in the synovial tissue of the OA group increased significantly compared to the control group. Moreover, synovial tissue in obese OA patients was infiltrated with more M1 macrophages than that in non-obese OA patients (*Figure 1B, D*). These results indicate that synovial tissues were highly hyperplastic in obese OA patients and infiltrated with more polarized M1 macrophages than in non-obese OA patients.

### Obesity promotes synovial M1 macrophage accumulation, synovitis, and OA development in mice

The *Apoe*$^{-/-}$ mouse model was established to further explore the role of obesity in OA development, as it is considered an ideal model for investigating obesity. Previous studies have shown that feeding high-fat diets to *Apoe*$^{-/-}$ mice for a short period accelerate the increase in LDL cholesterol levels and induce an inflammatory state (*Tung et al., 2020*; *Cao et al., 2020*). *Apoe*$^{-/-}$ mice may be clinically relevant to pathological progression in obese OA patients characterized by elevated plasma LDL cholesterol levels (*Gierman et al., 2014*; *Gierman et al., 2012*). The body weight and plasma lipid levels were markedly elevated in *Apoe*$^{-/-}$ mice after administering a high-fat and high-energy diet (*Tables 2–4*). There were no significant differences in knee OA OARSI scores between *Apoe*$^{-/-}$ and C57BL/6 mice 4 weeks post-surgery (*Figure 2—figure supplement 1*). However, the OARSI score was significantly elevated in *Apoe*$^{-/-}$ mice 8 weeks post-surgery (*Figure 2A, B*), accompanied by a higher synovitis score and more infiltrated inflammatory cells (*Figure 2A, C*), indicating that obesity may promote OA development in mice. In addition, *Apoe*$^{-/-}$ OA mice expressed less aggrecan on cartilage and more MMP13 on cartilage and synovium than C57BL/6 mice (*Figure 2D, E*). Notably, the percentage of M1-like macrophages was increased with OA progression and reached a maximum in obese *Apoe*$^{-/-}$ OA mice 8 weeks post-surgery. However, the proportion of positive cells for M2-like macrophages in the OA synovium showed no significant change at both 4 and 8 weeks post-surgery (*Figure 2F, G*, *Figure 2—figure supplement 2*). These findings suggest that obesity exacerbates synovitis and M1-polarized macrophage accumulation during OA progression in mice.

### GAS6 expression is inhibited in synovial macrophages during obesity-associated OA development

The link between M1 macrophages and synovial hyperplasia during obesity-associated OA progression was further explored. GAS6, a member of the vitamin K-dependent protein family, has been previously found to be down-regulated in Lipopolysaccharide (LPS)-induced bone marrow-derived macrophages (BMDMs) compared to the controls (GSE53986, *Figure 3—figure supplement 1*; *Noubade et al., 2014*). GAS6 was investigated as a critical factor regulating cell proliferation and apoptosis by binding to its receptor Axl. The GAS6/Axl effects on OA remain unclear. The present study explored the association between synovial macrophage polarization types and the GAS6/Axl pathway in obesity-associated OA. Immunofluorescence (IF) staining analysis indicated that macrophage release of GAS6 expressed in both human and mouse normal synovial tissues tended to be diminished, especially in obesity-associated OA (*Figure 3A–D*). Moreover, enzyme-linked immunosorbent assay (ELISA) revealed a significant decrease in GAS6 levels in the synovial fluid from obese

OA patients than in non-obese individuals (*Figure 3E*). An in vitro study in polarized M1 macrophages enhanced by LPS confirmed the M1 macrophage-associated reduction of GAS6 and increases in IL-1, IL-6, and TNF-α (*Figure 3F, G*, *Figure 3—figure supplement 2*, *Figure 3—figure supplement 2—source data 1*). Interestingly, F4/80-Axl double staining revealed no significant difference in the number of Axl-positive cells in the synovium of obese OA patients, obese *Apoe*$^{-/-}$ OA mice, and control subjects (*Figure 3—figure supplement 3*). These results indicate a potential role of M1 macrophage-mediated GAS6 in obesity-associated OA development.

## GAS6 is involved in obesity-mediated inhibition of macrophage efferocytosis

Efferocytosis is an indispensable process through which dead and dying cells are removed by phagocytic cells. MER were widely used to label macrophages undergoing efferocytosis (*Dransfield et al., 2015*; *Felton et al., 2018*; *Cai et al., 2017*). IF staining analysis indicated the expression of MER in OA synovial macrophages decreased significantly, especially in obese *Apoe*$^{-/-}$ OA and obese OA patients (*Figure 4—figure supplement 1*). Immunochemical TUNEL and caspase-3 staining in the present study revealed that the number of ACs was increased in synovial tissues with OA progression, which increased significantly in obese *Apoe*$^{-/-}$ OA and obese OA patients (*Figure 4A–D*). Previous studies have shown that efferocytosis of ACs induced by macrophages is impaired in inflammatory diseases. Still, its role in obesity-associated OA and understanding of its mechanism are lacking. In addition, GAS6 has been described as a crucial bridging protein for macrophages to recognize and engulf ACs. Therefore, it was hypothesized that the high percentage of observed ACs might be due to ineffective macrophage efferocytosis caused by GAS6 suppression in OA. Primary BMDMs from *Apoe*$^{-/-}$ mice and controls were fed carboxyfluorescein succinimidyl ester (CFSE)-labeled ACs to determine the potency of efferocytosis induced by macrophages. The clearance of these fluorescent cells was quantified using flow cytometry. As expected, macrophage efferocytosis was impaired in the obesity microenvironment with reduced capacity for clearing fluorescent ACs in BMDMs from *Apoe*$^{-/-}$ mice compared to the controls (*Figure 4E, F*). Moreover, stimulation with GAS6 enhanced the phagocytotic activity of RAW264.7 cells, while inhibition of the Axl receptor by adding R428 diminished this effect (*Figure 4H–K*), which were further verified in BMDM primary cells (*Figure 4—figure supplement 2*). Adding recombinant mouse GAS6 (rmGAS6) significantly restored the up-regulation of inflammatory factors IL-1β, IL-6, and TNF-α induced by LPS in RAW264.7 cells (*Figure 3—figure supplement 2*). Nevertheless, rmGAS6 had no obvious effect on the polarization of macrophages, while R428 up-regulated CD86 and iNOS in BMDMs (*Figure 4G*, *Figure 4—figure supplement 3*). These data indicate that M1 macrophage-associated reduction of GAS6 in obesity-associated OA mice promotes the accumulation of ACs by decreasing macrophage efferocytosis.

## Suppression of GAS6/Axl axis promotes synovial hyperplasia, synovitis, and obesity-associated OA development

To further investigate the role of GAS6/Axl signaling in the development of OA in vivo, an intra-articular injection intervention was performed in OA mice. As a result, the degree of synovial inflammation and cartilage degeneration in C57BL/6 mice was far lower than that in *Apoe*$^{-/-}$ mice, with a lower level of GAS6 expression in macrophages (*Figure 2A–C and 3C, D*). Therefore, rmGAS6 was injected into *Apoe*$^{-/-}$ obese OA mice to protect against synovial hyperplasia and cartilage damage

**Table 2.** Comparison of specifications and energy of ordinary feed and high-fat feed.

| Composition | Ordinary feed | | High-fat feed | |
|---|---|---|---|---|
| | g (%) | kcal (%) | g (%) | kcal (%) |
| Protein | 19.2 | 20 | 24 | 20 |
| Carbohydrate | 67.3 | 70 | 41 | 35 |
| Fat | 4.3 | 10 | 24 | 45 |
| Total | | 100 | | 100 |
| kcal/g | 3.85 | | 4.73 | |

**Table 3.** Lipid status of APOE⁻/⁻ obese mice and C57BL/6 mice.

| | *Apoe*⁻/⁻ mice | C57BL/6 mice |
|---|---|---|
| Total cholesterol (TC) (3.0–6.0) | 17.82 ± 3.3 | 2.77 ± 0.62 |
| Triglyceride (TG) (0.56–1.7) | 2.56 ± 0.43 | 0.92 ± 0.31 |

induced by GAS6/Axl pathway suppression. On the other hand, the inhibitor R428 was injected into the joint cavity of C57BL/6 OA mice to stimulate OA progression. Interestingly, intra-articular injection of GAS6 significantly delayed synovial inflammation and cartilage destruction compared to vehicle-treated obese OA mice, manifested as lower synovitis and OARSI scores. In contrast, inhibition of Axl by R428 in C57BL/6 OA mice promoted synovial hyperplasia and cartilage destruction and enhanced synovitis and OARSI scores (*Figure 5A, B*). Moreover, intra-articular injection of GAS6 recombinant factor decreased the number of ACs stimulated by synovitis in synovial tissues (*Figure 5C, D*). To further explore the effect of inflammatory factors released by the stimulation of ACs on chondrocyte homeostasis dysfunction, ACs were co-cultured with BMDMs. The culture supernatant was collected after stimulation for 24 hr. The expression of MMP13 was increased after adding co-culture supernatant stimulation, together with senescence hallmarks such as p16 and p21, while COL2 were down-regulated after stimulation. However, the effect of supernatant promoting chondrocyte homeostasis dysfunction was partially alleviated when adding ACs with rmGAS6 to BMDMs, which were diminished by adding R428 (*Figure 5—figure supplement 1*, *Figure 5—figure supplement 1—source data 1*). Nevertheless, no apparent proteoglycan loss or increase was found in recombinant human GAS6 (rhGAS6)-treated cartilage explants (*Figure 5—figure supplement 2*). Moreover, western blotting showed no significant differences in the expression of COL2, MMP13, p16, and p21 in primary chondrocytes after stimulation with rhGAS6 (*Figure 5—figure supplement 3*, *Figure 5—figure supplement 3—source data 1*). Therefore, these data indicated that the GAS6/Axl axis might alleviate synovial hyperplasia and protect against obesity-associated OA development.

## Model for obesity-associated synovitis and OA development

Obesity stimulates synovial macrophage infiltration and M1 polarization, which suppress the secretion of GAS6. GAS6 binds to the Axl receptor on macrophages, while GAS6/Axl inhibition promotes the accumulation of ACs by decreasing macrophage efferocytosis to induce chondrocyte degradation and aggravate OA development (*Figure 6*).

## Discussion

The present study revealed for the first time that M1-polarized macrophage infiltration in OA synovial tissue of obese patients is increased, accompanied by markedly down-regulated secretion of GAS6 and impaired macrophage-dependent efferocytosis for cleaning ACs. The intracellular contents released by accumulated ACs further trigger an immune response and lead to a release of inflammatory factors, such as TNF-α, IL-1β, and IL-6, which induce the dysfunction of chondrocyte homeostasis in obesity-associated OA. Therefore, targeting macrophage-associated efferocytosis or intra-articular injection of GAS6 is a potential therapeutic strategy for obesity-associated OA.

Obesity has always been considered a significant risk factor in the progression of OA (*Kulkarni et al., 2016*), leading to more severe OA manifestations, including cartilage loss, subchondral bone sclerosis, and synovial inflammation (*Wang and He, 2018*). Researchers have accepted the role of synovial inflammation in the pathological progression of OA (*Mathiessen and Conaghan, 2017*; *Felson et al., 2016*). However, the underlying mechanism of obesity-related inflammation in the development of synovitis in OA remains unclear. *Apoe* plays an important role in maintaining the normal levels of cholesterol and triglycerides in serum by transporting lipids in the blood (*Hatters et al., 2006*). Mice lacking *Apoe* function develop hypercholesterolemia, increased very low-density

**Table 4.** Weight gain after feeding for 8 weeks (g).

| | C57BL/6 mice | *Apoe*⁻/⁻ mice |
|---|---|---|
| Standard diet | 7.35 ± 1.22 | 9.13 ± 0.78 |
| High-fat diet | 16.89 ± 0.75 | 19.81 ± 1.33 |

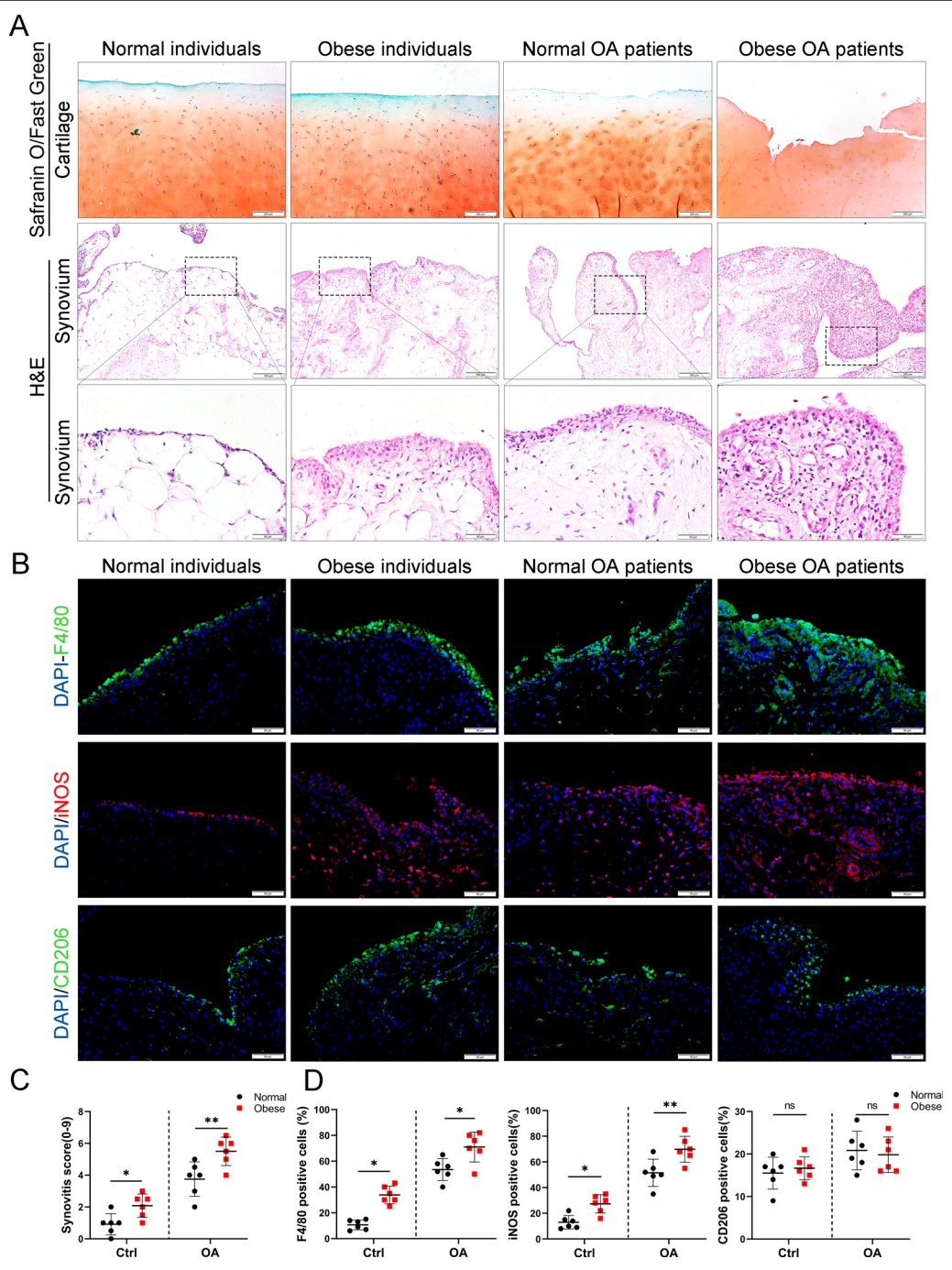

**Figure 1.** Synovial hyperplasia and macrophage polarization in obese OA patients. (**A**) Safranin O and Fast Green staining (top) of human articular cartilage, hematoxylin and eosin (H&E) staining (lower) of synovial tissue from normal individuals, OA patients without obesity, obese individuals, and OA patients with obesity. Scale bar: 200 µm, 50 µm. (**B**) Immunofluorescence of F4/80, inducible nitric oxide synthase (iNOS), and CD206 in normal and OA synovial tissues from normal and obese patients. F4/80: green; iNOS: red; DNA: blue. Scale bar: 50 µm. (**C**) Quantification of synovitis score in normal individuals, OA patients without obesity, obese individuals, and OA patients with obesity (*n* = 6 per group). (**D**) Quantification of F4/80, iNOS, and CD206 positive macrophages as a proportion of total lining cell population in (**B**). *p < 0.05, **p < 0.01, ns = not significant. One-way analysis of variance (ANOVA) was performed. Data are shown as mean ± standard deviation (SD).

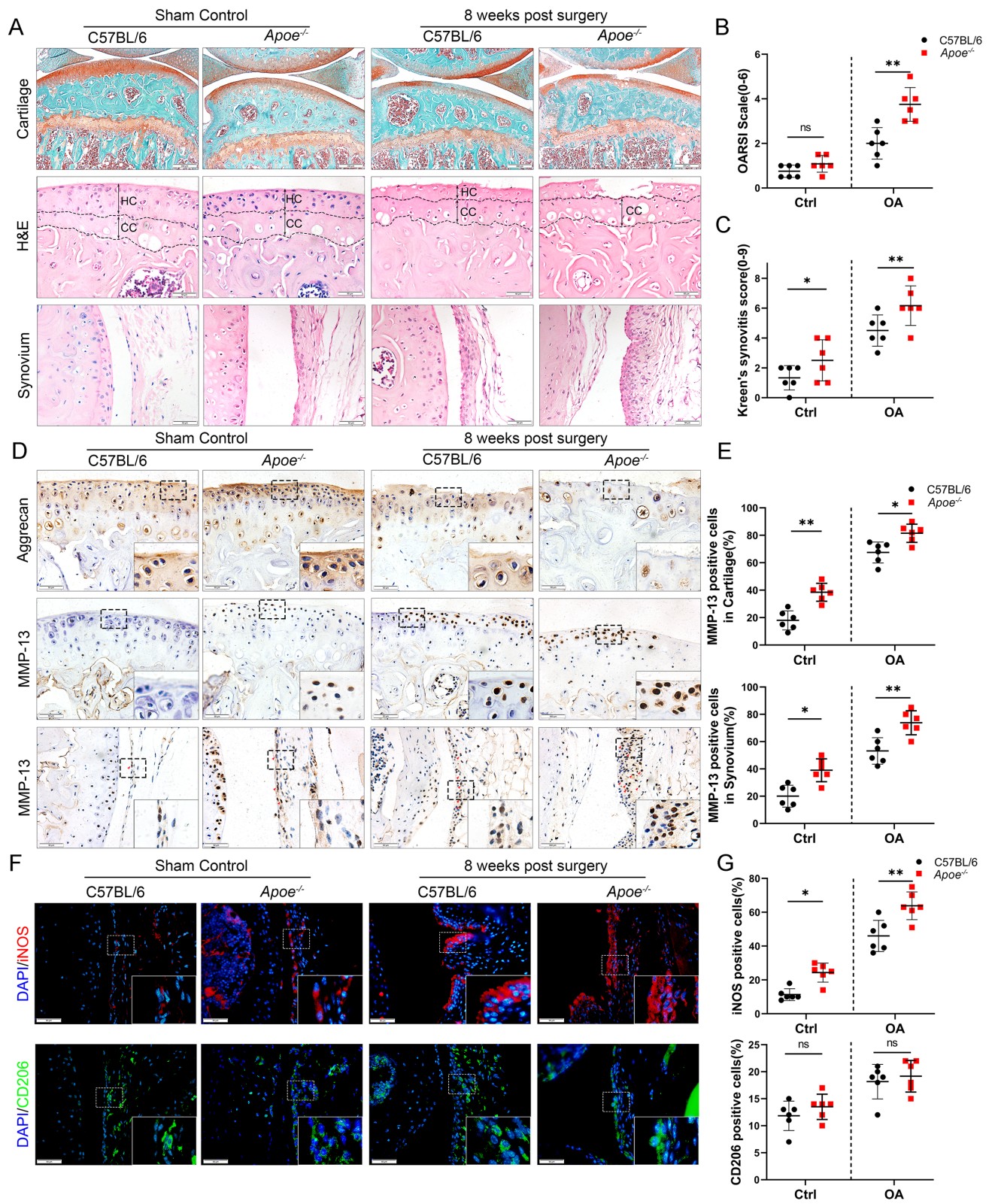

**Figure 2.** Cartilage loss, synovial hyperplasia, and macrophage polarization in *Apoe*[−/−] OA. (**A**) Safranin O and Fast Green (first line) and hematoxylin and eosin (H&E; second line) staining of controls and destabilization of medial meniscus (DMM) knee cartilage or synovial membrane from normal and *Apoe*[−/−] mice. Scale bar: 200 μm, 50 μm. (**B**) Quantitative analysis of Osteoarthritis Research Society International (OARSI) scale in A (second line), *n* = 6 per group. (**C**) Synovitis score for joints described in (**A**) (third line), *n* = 6 per group. (**D**) Immunohistochemical staining for aggrecan (first line) and MMP-

*Figure 2 continued on next page*

*Figure 2 continued*

13 (middle and bottom) in controls and DMM knee cartilage from normal and *Apoe*⁻/⁻ mice. Scale bar: 50 μm. (**E**) Quantification of MMP13-positive cells from cartilage or synovium in (**D**), n = 6 per group. (**F**) Immunofluorescence staining for inducible nitric oxide synthase (iNOS; first line) and CD206 (second line) in controls and DMM synovial tissues from normal and *Apoe*⁻/⁻ mice. Scale bar: 50 μm; (**G**) Quantification of iNOS- and CD206-positive cells as a proportion of lining cell population in (**F**), n = 6 per group. *p < 0.05, **p < 0.01, ns = not significant. One-way analysis of variance (ANOVA) was performed. Data are shown as mean ± standard deviation (SD).

The online version of this article includes the following figure supplement(s) for figure 2:

**Figure supplement 1.** Cartilage loss and synovial hyperplasia in 4 week-old Apoe⁻/⁻ OA mice.

**Figure supplement 2.** Macrophage polarization in 4 week-old Apoe⁻/⁻ OA mice.

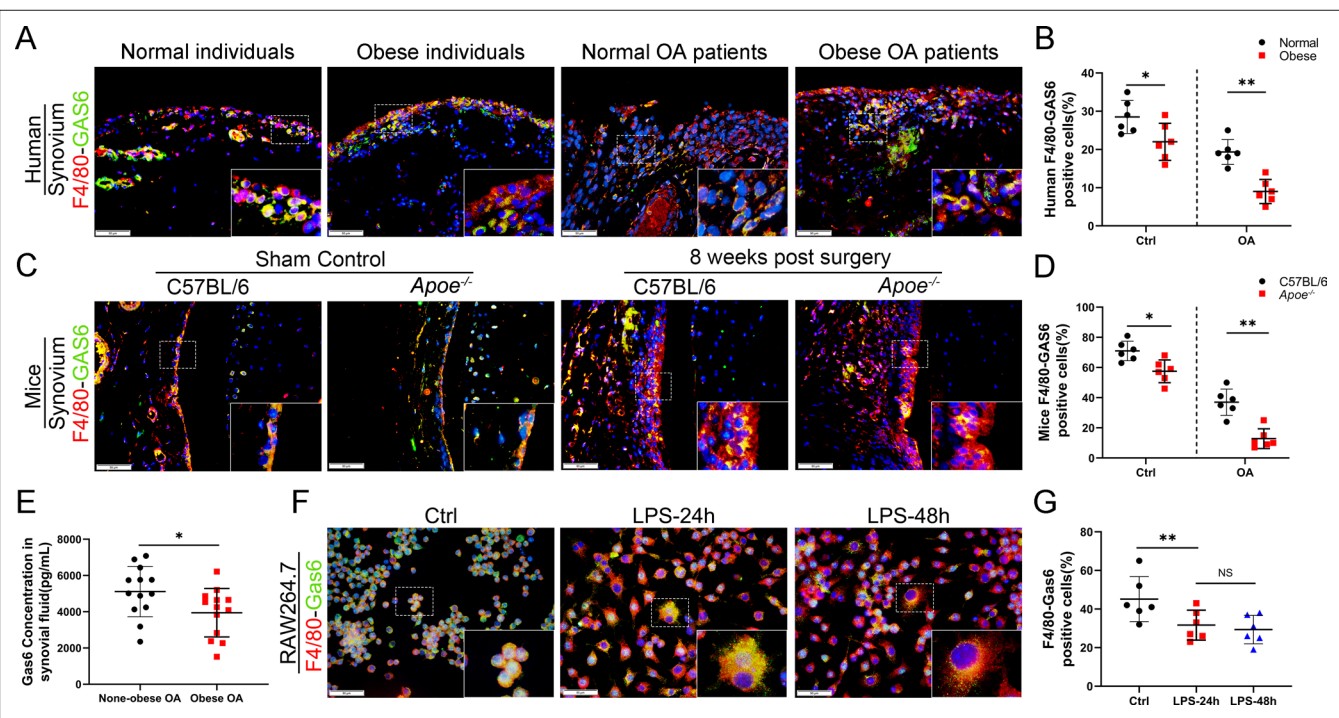

**Figure 3.** Loss of GAS6 expression in synovium of obese OA patients and *Apoe*⁻/⁻ OA mice. (**A**) Immunofluorescence staining for F4/80 (red) and GAS6 (green) in synovial tissue from normal individuals, OA patients without obesity, obese individuals, and OA patients with obesity. Scale bar: 50 μm. (**B**) Quantification of F4/80-GAS6-positive macrophages as a proportion of total lining cell population in (**A**), n = 6 per group. (**C**) Immunofluorescence staining (first line) for F4/80 (red) and GAS6 (green) in synovial tissue of controls and destabilization of medial meniscus (DMM) from C57BL/6 and *Apoe*⁻/⁻ mice. Scale bar: 50 μm. (**D**) Quantification of F4/80-GAS6-positive macrophages (yellow) as a proportion of total F4/80-positive cells in (**C**) (first line). Quantification of GAS6-positive cells in (**C**) (second line), n = 6 per group. (**E**) Enzyme-linked immunosorbent assay (ELISA) for GAS6 in synovial fluid of non-obese and obese OA patients, n = 13 per group. (**F**) Immunofluorescence staining for F4/80(red) and GAS6 (green) in RAW264.7 cells treated with LPS for 24 and 48 hr. Scale bar: 50 μm. (**G**) Quantification of F4/80-GAS6-positive macrophages (yellow) as a proportion of total F4/80-positive cells (red), n = 6 per group. *p < 0.05, **p < 0.01, NS = not significant. One-way analysis of variance (ANOVA) was performed. Data are shown as mean ± standard deviation (SD).

The online version of this article includes the following source data and figure supplement(s) for figure 3:

**Figure supplement 1.** Differentially expressed mRNA in bone marrow-derived macrophages from normal controls or LPS treatment based on GSE53986.

**Figure supplement 1—source data 1.** Results of GSE53986.

**Figure supplement 2.** The expression of GAS6 and inflammatory cytokines in M1/M2 polarized macrophages.

**Figure supplement 2—source data 1.** Primary blots related to *Figure 3—figure supplement 2*.

**Figure supplement 3.** AXL expression in synovium of obese OA patients and Apoe⁻/⁻ OA mice.

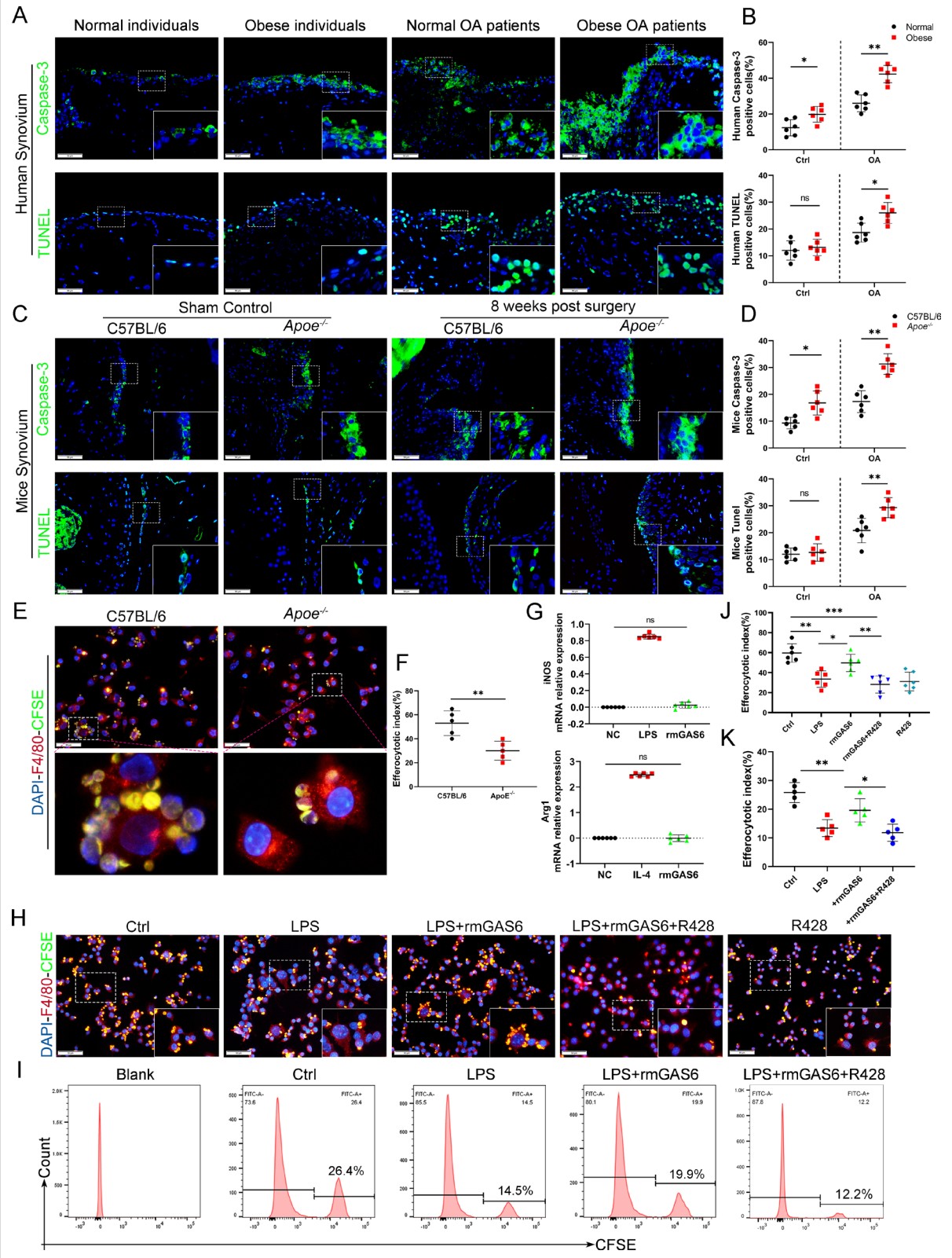

**Figure 4.** Accumulation of apoptotic cells in OA and impaired phagocytic ability of M1-polarized macrophages. (**A**) Immunofluorescence staining for caspase-3 (top) and TUNEL (lower) in normal and OA synovial tissue from non-obese and obese patients. Scale bar: 50 μm. (**B**) Quantification of caspase-3- or TUNEL-positive cells as a proportion of total lining cell population in (**A**), *n* = 6 per group. (**C**) Immunofluorescence staining for caspase-3 (top) and TUNEL (lower) in controls and destabilization of medial meniscus (DMM) synovial tissue from C57BL/6 and *Apoe*[−/−] mice. Scale bar: 50 μm. (**D**)

*Figure 4 continued on next page*

*Figure 4 continued*

Quantification of caspase-3- or TUNEL-positive cells as a proportion of lining cell population in (**C**), *n* = 6 per group. (**E**) Immunofluorescence staining for F4/80 (red) in bone marrow-derived macrophages (BMDMs) extracted from *Apoe*⁻/⁻ and C57BL/6 mice. Carboxyfluorescein succinimidyl ester (CFSE; green) in apoptotic thymocytes of C57BL/6 mice after 2 hr phagocytosis. (**F**) Quantification of positive BMDMs engulfing apoptotic thymocytes as a proportion of total F4/80-positive cells, *n* = 5 per group. (**G**) mRNA expression levels of inducible nitric oxide synthase (iNOS) or Arg1 after LPS, rmGAS6, or IL-4 stimulation for 24 hr. (**H**) Immunofluorescence staining for F4/80 (red) in RAW264.7 cells and CFSE (green) in apoptotic thymocytes after phagocytosis for 2 hr. Scale bar: 50 μm. (**I**) Flow cytometry analysis of CFSE-positive cells in total macrophages is shown as fluorescence-intensity distribution plots. (**J**) Quantification of positive RAW264.7 cells engulfing apoptotic thymocytes as a proportion of total F4/80-positive cells, *n* = 6 per group. (**K**) Efferocytotic index was calculated as percentage of CFSE-positive cells divided by percentage of total cells, *n* = 5 per group. *p < 0.05, **p < 0.01, ***p < 0.001, NS = not significant. One-way analysis of variance (ANOVA) was performed. Data are shown as mean ± standard deviation (SD).

The online version of this article includes the following source data and figure supplement(s) for figure 4:

**Figure supplement 1.** The expression of MER decreased in obese OA patients and Apoe⁻/⁻ OA mice.

**Figure supplement 2.** rmGAS6 attenuated the impaired efferocytosis induced by LPS.

**Figure supplement 3.** Western blot of CD86, inducible nitric oxide synthase (iNOS) in RAW264.7 cells after R428 stimulation.

**Figure supplement 3—source data 1.** Primary blots related to *Figure 4—figure supplement 3*.

lipoprotein and decreased high-density lipoprotein, leading chronic inflammation in vascular disease and nonalcoholic steatohepatitis (*Liu et al., 2023*). *Apoe*⁻/⁻ mice may partially reflect clinical characteristics of obese OA patients with elevated plasma LDL cholesterol levels. The present study revealed that obese patients and *Apoe*⁻/⁻ mice showed a more severe cartilage destruction with enhanced OARSI scores (*Glasson et al., 2010*). Furthermore, the lining layer of synovial tissue was more prone to hyperplasia. The number of macrophages increased significantly during the pathological development of OA in obese patients, which manifested as enhanced synovitis scores (*Krenn et al., 2006*) and increased numbers of F4/80-positive cells. These results suggest that obesity plays an essential mediating role in OA development.

Recent studies have reported that the imbalance in M1/M2 macrophage polarization plays a vital role in developing OA inflammation (*Zhang et al., 2018*). Classically, macrophages are divided into inflammatory M1 and anti-inflammatory M2 macrophages, though they can be interchanged or transformed into each other during various inflammatory reactions and thus function differently (*Funes et al., 2018*). *Lumeng et al., 2007* have found that adipose tissue macrophages (ATMs) prefer to express TNF-α, iNOS, and other M1 macrophage markers, while ATMs in non-obese individuals highly express M2 macrophage markers. However, the effect of obesity on inducing macrophage polarization during OA development remains unclear. Many bursas and adipose tissue around the knee joint are typically characterized as the infrapatellar fat pad. Some researchers believe that the transformation of macrophages is triggered by lipids released by fat cells at the onset of obesity (*Chatterjee et al., 2013*). Thus, we investigated whether macrophage abnormalities in obesity-related knee OA affect the synovial membrane in the knee joint. We found that M1 macrophage infiltration increased in the OA synovial tissue of obese patients and *Apoe*⁻/⁻ mice compared to non-obese patients and C57BL/6 mice, accompanied by increased secretion of TNF-α, IL-1, and IL-6. These results suggest that obesity may be a crucial factor affecting the functional status of macrophages, and targeting the polarization of macrophages in obese patients may alleviate the synovitis in OA.

The balance of progression and regression that controls the state of local inflammation has recently been in the spotlight (*Oishi and Manabe, 2018*; *Feehan and Gilroy, 2019*). The maintenance of efferocytosis dampens pro-inflammatory cytokine production and initiates inflammation resolution (*Boada-Romero et al., 2020*). Emerging studies have mentioned the effect of macrophage polarization on phagocytic ability. Yurdagul et al. have shown that M2 macrophages retain efferocytosis properties, which are crucial for resolving inflammation (*Yurdagul et al., 2020*). Another study demonstrated that DEL-1 contributes to the resolution of inflammation by promoting apoptotic neutrophil efferocytosis through macrophages and the emergence of an M2 macrophage phenotype (*Kourtzelis et al., 2019*). Although efferocytosis has been widely studied in multiple disease models, its role in synovial inflammation and OA development has not been reported. We found that the expression of MER decreased significantly in OA synovial macrophages, especially in obesity-associated OA. Our in vitro experiments demonstrated that macrophages extracted from the bone marrow of obese mice had decreased phagocytic capacity, and the phagocytic ability of macrophages for apoptotic thymocytes was reduced after stimulation with LPS (500 ng/ml). These results showed that the proportion of

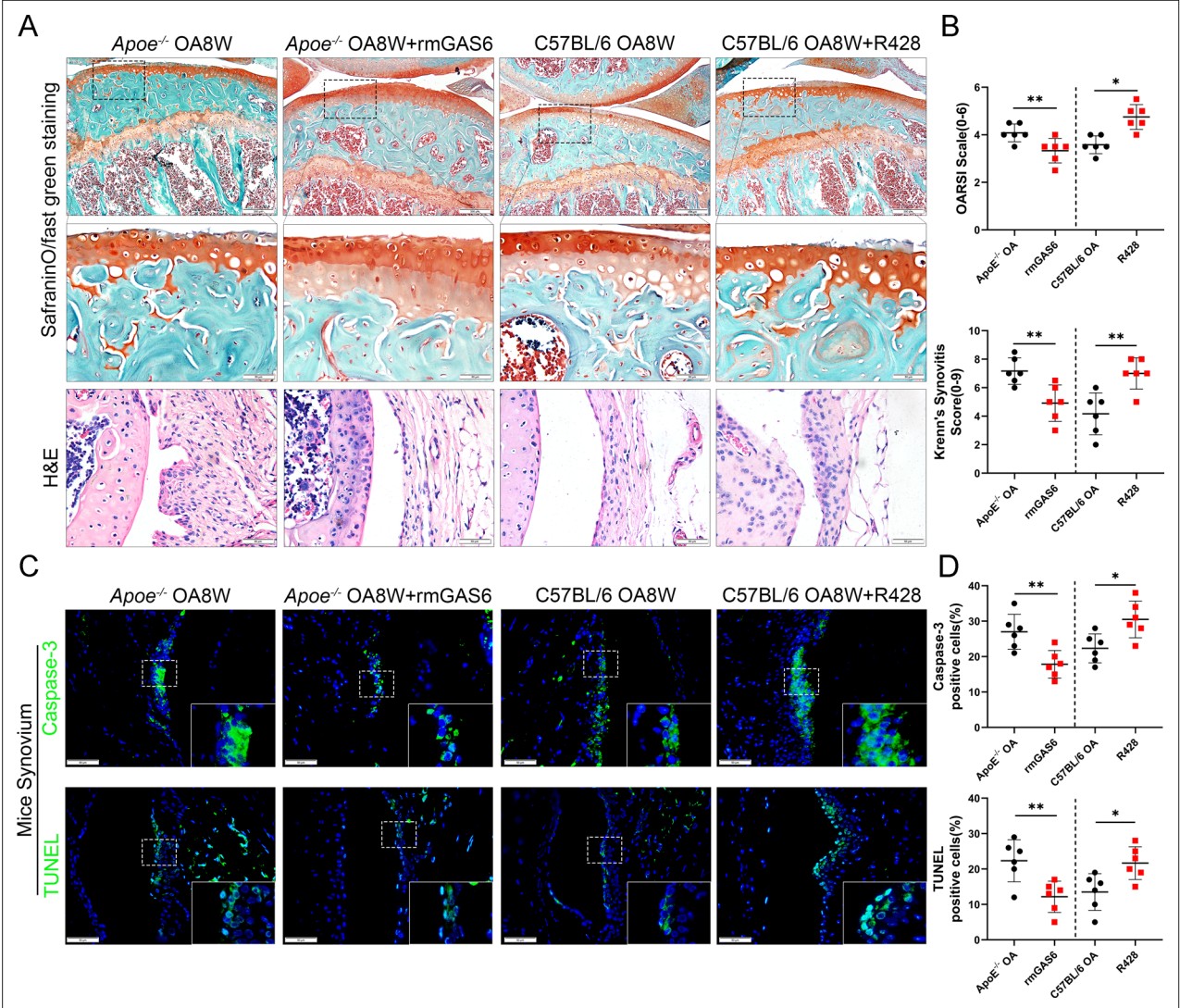

**Figure 5.** GAS6 restored osteoarthritis (OA) cartilage loss and decreased apoptotic cell accumulation. (**A**) Safranin O and Fast Green staining (top and middle) of knee cartilage, hematoxylin and eosin (H&E) staining of synovial tissues from destabilization of medial meniscus (DMM) mice and DMM mice treated with R428, and $Apoe^{-/-}$ mice treated with recombinant mouse (rmGAS6) 8 weeks after surgery. Scale bar: 200 μm, 50 μm. (**B**) Quantitative analysis of Osteoarthritis Research Society International (OARSI) scale and synovitis score in (**A**), n = 6 per group. (**C**) Immunofluorescence staining of caspase-3 or TUNEL in synovial tissue from DMM mice, DMM mice treated with R428, and $Apoe^{-/-}$ mice treated with recombinant mouse (rmGAS6) 8 weeks after surgery. Scale bar: 50 μm. (**D**) Quantification of caspase-3- or TUNEL-positive cells as a proportion of lining cell population in (**C**), n = 6 per group. *p < 0.05, **p < 0.01. One-way analysis of variance (ANOVA) was performed. Data are shown as mean ± standard deviation (SD).

The online version of this article includes the following source data and figure supplement(s) for figure 5:

**Figure supplement 1.** Immunoblotting and quantification of MMP13, COL2, and senescence markers (p16, p21) in primary chondrocytes treated with supernatant from bone marrow-derived macrophages (BMDMs) which co-cultured with apoptotic cells (ACs) or rmGAS6 and R428.

**Figure supplement 1—source data 1.** Primary blots related to *Figure 5—figure supplement 1*.

**Figure supplement 2.** Safranin O staining of human tibial plateaus cartilage explants treated with rhGAS6.

**Figure supplement 3.** Immunoblotting of MMP13, COL2, and senescence markers (p16, p21) in primary chondrocytes treated with rhGAS6.

**Figure supplement 3—source data 1.** Primary blots related to *Figure 5—figure supplement 3*.

ACs was significantly increased in the synovium of $Apoe^{-/-}$ mice. This is partly due to the decreased phagocytic ability caused by the enhanced M1 macrophage polarization. We have further revealed that macrophages stimulated by ACs leading chondrocytes homeostasis dysfunction, yet administration of rmGAS6 partially alleviated the up-regulation of p16, p21, and MMP13. Intra-articular injection of GAS6 restored the phagocytic capacity of macrophages. It reduced the accumulation of local

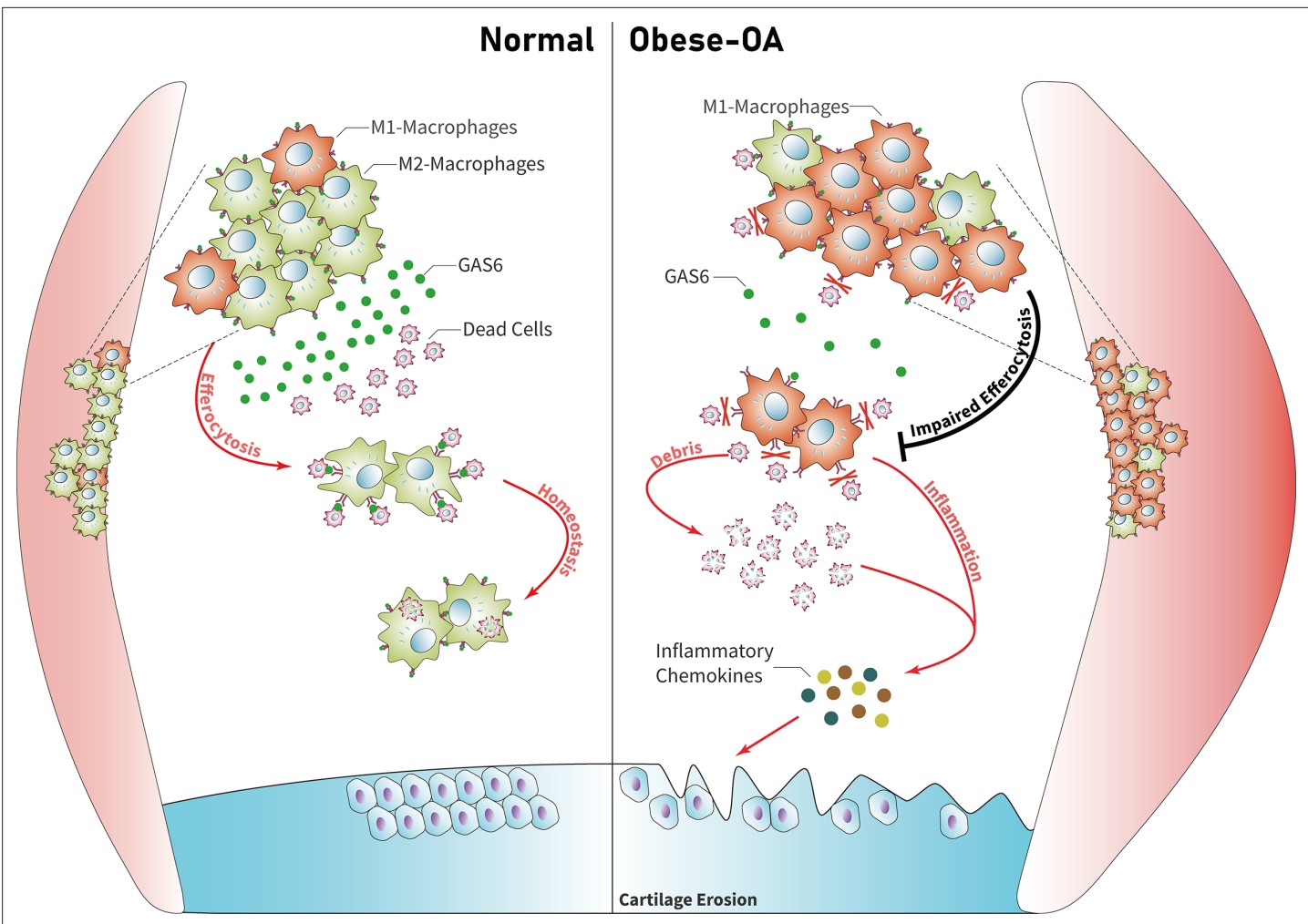

**Figure 6.** Model of GAS6 secreted by macrophages in modulating clearance of apoptotic cells and macrophage polarization during osteoarthritis (OA). Macrophage polarization induced by obesity decreased the secretion of GAS6 and impaired the phagocytosis of apoptotic cells. The accumulation of apoptotic cell debris leads to the persistence of local inflammation and synovial hyperplasia, which aggravates the pathological process of OA.

ACs and decreased the levels of TUNEL and caspase-3-positive cells, preserving cartilage thickness and preventing the progression of obesity-associated OA. These findings suggest that targeting the efferocytosis of macrophages in local inflammation of obesity-associated synovitis may maintain the homeostasis of the cartilage cavity and alleviate the obesity-related OA.

GAS6 and its receptor Axl are known to regulate apoptotic-related inflammation and may be implicated in lupus pathogenesis (*Zhen et al., 2018*). The remaining ACs are a source of autoantigens and can drive autoimmunity development (*Szondy et al., 2014*). The expression of GAS6 in the hyperplasia synovial tissues from obesity-associated OA in the present study was down-regulated and accompanied by an increase in M1 macrophage polarization. Moreover, the in vitro experiments demonstrated a decreased secretion of GAS6 protein and impaired phagocytic ability in macrophages after LPS stimulation. However, the proportion of macrophages that engulfed ACs was increased after incubation with recombinant GAS6 protein, while the phagocytic ability was significantly down-regulated after blocking the Axl receptor by adding R428. Exogenous cultured rmGAS6 with macrophages after stimulation of LPS can also decrease the levels of inflammatory cytokines such as IL-1, IL-6, and TNF-α. Nevertheless, there was no significant difference in the expression of its specific receptor Axl, which may be explained by the fact that GAS6 has three receptors (Axl, Mer, and Tyro3) with different affinities. These findings suggest that GAS6 may relieve local synovial inflammation by restoring the phagocytic ability of macrophages for ACs and decrease the induction of inflammatory chemokines, which alleviate the pathological progression of OA.

To conclude, the present study found that obese OA patients and *Apoe*$^{-/-}$ obese mice showed a more pronounced synovitis and enhanced macrophage infiltration in synovial tissue, accompanied by dominant M1 macrophage polarization. Obese OA mice had more severe cartilage destruction than OA mice in the control group. Enhanced M1-polarized macrophages in obese synovium decreased GAS6 secretion, impairing efferocytosis for synovial ACs and causing synovial hyperplasia and obesity-associated OA development. Therefore, these findings reveal that targeting GAS6-mediated macrophage polarization and phagocytosis in obese patients with OA may be a potential therapeutic strategy.

# Materials and methods

## Key resources table

| Reagent type (species) or resource | Designation | Source or reference | Identifiers | Additional information |
|---|---|---|---|---|
| Cell line (*Mus musculus*) | RAW264.7 | Pricella | HC2022083021 | Cell line has been authenticated by STR profiling and it did not be contaminated by mycoplasma |
| Antibody | anti-F4/80 (Mouse monoclonal) | Santa Cruz Biotechnology | Cat #: sc377009 | IF (1:100) |
| Antibody | anti-iNOS (Mouse monoclonal) | Santa Cruz Biotechnology | Cat #: sc-7271 | IF (1:100) |
| Antibody | anti-Aggrecan (Rabbit polyclonal) | Proteintech | Cat #: 13880-1-AP | IF (1:200) |
| Antibody | anti-MMP13 (Rabbit polyclonal) | Proteintech | Cat #: 18165-1-AP | IF (1:400) WB (1:1000) |
| Antibody | anti-CD206 (Rabbit polyclonal) | Proteintech | Cat #: 18704-1-AP | IF (1:100) |
| Antibody | anti-AXL (Rabbit polyclonal) | Abclone | Cat #: A20548 | IF (1:100) |
| Antibody | anti-CASPASE-3 (Rabbit polyclonal) | Proteintech | Cat #: 19677-1-AP | IF (1:200) |
| Antibody | anti-GAS6 (Rabbit polyclonal) | Abclone | Cat #: A8545 | IF (1:100) |
| Antibody | Peroxidase AffiniPure Goat Anti-Rabb (Goat polyclonal) | Jackson Immuno Research Laboratories | Cat #: 11-035-003 | IHC (1:200) WB (1:3000) |
| Antibody | Goat anti-Rabbit IgG (H+L) Cross-Adsorbed Secondary Antibody, Alexa Fluo 488 (Goat polyclonal) | Invitrogen | Cat #: A-11008 | IF (1:400) |
| Antibody | Goat anti-Mouse IgG (H+L) Cross-Adsorbed Secondary Antibody, Alexa Fluor 594 (Goat polyclonal) | Invitrogen | Cat #: A-11005 | IF (1:400) |
| Antibody | Anti-Collagen II antibody (Rabbit polyclonal) | Abcam | Cat #: ab188570 | WB (1:1000) |
| Sequence-based reagent | *Gas6*_F | This paper | PCR primers | CCGCGCCTACCAAGTCTTC |
| Sequence-based reagent | *Gas6*_R | This paper | PCR primers | CGGGGTCGTTCTCGAACAC |
| Sequence-based reagent | Gapdh _F | This paper | PCR primers | AAATGGTGAAGGTCGGTGTGAAC |
| Sequence-based reagent | Gapdh _R | This paper | PCR primers | CAACAATCTCCACTTTGCCACTG |
| Sequence-based reagent | *Il-1β*_F | This paper | PCR primers | GCAACTGTTCCTGAACTCAACT |
| Sequence-based reagent | *Il-1β*_R | This paper | PCR primers | ATCTTTTGGGGTCCGTCAACT |
| Sequence-based reagent | *Il-6*_F | This paper | PCR primers | ACAACCACGGCCTTCCCTACTT |
| Sequence-based reagent | *Il-6*_R | This paper | PCR primers | CAGGATTTCCCAGCGAACATGTG |
| Sequence-based reagent | *Tnf-α*_F | This paper | PCR primers | CCTCCCTCTCATCAGTTCTA |
| Sequence-based reagent | *Tnf-α*_R | This paper | PCR primers | ACTTGGTTTGCTACGAC |
| Commercial assay or kit | TUNEL Apoptosis Detection Kit (Alexa Fluor 488) | Yeasen | Cat #: 40307ES20 | - |
| Commercial assay or kit | Human Gas6 DuoSet ELISA | R&D | Cat #: DY885B | - |
| Commercial assay or kit | RNAiso Plus (Trizol) | Takara Bio Inc | Cat #: T9108 | - |
| Commercial assay or kit | 5× HiScript II qRT SuperMix II | Vazyme Biotech | Cat #: R223-01 | - |
| Commercial assay or kit | 2× ChamQ SYBR qPCR Master Mix | Vazyme Biotech | Cat #: Q311-02 | - |
| Chemical compound, drug | DAPI | Sigma-Aldrich | Cat #: F6057-20ML | - |
| Chemical compound, drug | Carboxyfluorescein succinimidyl ester (CFSE) | Topscience | Cat #: T6802 | - |

*Continued on next page*

*Continued*

| Reagent type (species) or resource | Designation | Source or reference | Identifiers | Additional information |
|---|---|---|---|---|
| Peptide, recombinant protein | R428 | Topscience | Cat #: 1037624-75-1 | - |
| Peptide, recombinant protein | Lipopolysaccharide | Med Chem Express | Cat #: HY-D1056 | - |
| Peptide, recombinant protein | IL-4 Protein, Mouse (CHO) | Med Chem Express | Cat #: HY-D1056 | - |
| Peptide, recombinant protein | Recombinant GAS6 Protein (Mouse) | Sino Biological | Cat #: 58026-M08H | - |
| Peptide, recombinant protein | Recombinant GAS6 Protein (Human) | Novoprotein | Cat #: C01W | - |
| Software, algorithm | SPSS | SPSS | SPSS 25.0 | - |

## Human synovial and cartilage tissue

Synovial tissue and synovial fluid samples of normal individuals ($n = 6$, age $34 \pm 8.15$ years, three males, three females) or obese individuals ($n = 6$, age $35 \pm 7.36$ years, four males, two females) were obtained from patients who received arthroscopic treatment for acute anterior cruciate ligament rupture or meniscus injury Other joint diseases were excluded from the study. OA synovial tissues, and synovial fluid samples were obtained from obese patients ($n = 6$, age $64 \pm 5.75$ years, two males, four females) or patients without obesity ($n = 6$, age $65 \pm 4.26$ years, three males, three females) who underwent total knee arthroplasty. OA cartilage tissues were obtained from patients who underwent total knee arthroplasty, the tibial plateau cartilage was carefully separated and cut into 1 mm$^3$, cultured in vitro. Informed consent was obtained from all recruited patients and was identified by the ethics committee of the Third Affiliated Hospital of Southern Medical University.

## Destabilization of medial meniscus animal model

Ten-week-old male C57BL/6 mice and *Apoe*-deficient (*Apoe*$^{-/-}$) male mice were purchased from the Experimental Animal Center of Guangdong Province, China. All animals were housed in cages without pathogens at a temperature of $24 \pm 5°C$ and with a relative humidity of 40%. C57BL/6 mice were fed a standard diet, and *Apoe*$^{-/-}$ mice were fed a high-fat diet. The feed specifications are shown in *Table 2*. This study was performed in strict accordance with the recommendations in Chinese Laboratory animal-Guideline for ethical review of animal welfare (GB/T 35892-2018). The protocol was approved by the Southern Medical University Animal Care and Use Review Board (Permit Number: 2021-Ethical review-053). All surgery was performed under sodium pentobarbital anesthesia, and every effort was made to minimize suffering.

In the destabilization of medial meniscus-OA model (*Kamekura et al., 2005*), mice were anesthetized by intraperitoneal injection of 0.3% sodium pentobarbital and the skin was cut along the medial collateral ligament. The joint capsule was cut open and the femoral condyle was exposed. The connection between the medial meniscus and the tibial plateau was cut to release the medial meniscus. The joint capsule and skin were sutured after the operation.

## Animal treatment and specimen preparation

After the surgery, 50 ng/g (about 5 µl as total volume) of recombinant mouse GAS6 (rmGAS6, Sino Biological, China, #58026-M08H) was administered into the articular once per week. The right legs were harvested 4–8 weeks post-surgery ($n = 6$ in each group). Knee joints from mice in different experimental groups were fixed in 4% paraformaldehyde for 48 hr and decalcified for 21 days. The specimens were embedded in paraffin, and 4 µm serial sections were cut from the sagittal portion through the inner side of the knee. The Southern Medical University Animal Care and Use Committee approved all procedures involving mice.

## Histology and immunohistochemical/IF staining

Histology sections were stained with Safranin O-fast green/hematoxylin and eosin for morphological analysis. Immunohistochemical (IHC) and IF staining was performed on the 4-µm-thick tissue sections. Slides were deparaffinized, rehydrated, and washed in phosphate-buffered saline (PBS)

three times for 5 min each time. Antigen retrieval was performed by soaking slides in citric acid over-night in a 60°C water bath. After washing three times in PBS, slides were quenched in 3% hydrogen peroxide for 10 min at room temperature and washed with PBS three more times. Then, slides were blocked with 10% normal bovine serum for 1 hr at room temperature (IHC staining). Slides were then incubated with primary antibodies at 4°C overnight. A secondary antibody for IHC or fluorescent secondary antibody for IF was applied for 1 hr at room temperature. Then, IHC slides were stained with diaminobenzidine and hematoxylin, dehydrated, and mounted. IF slides were processed with 4,6-diamidino-2-phenylindole (DAPI) staining solution and mounted with cover glass. Antibodies used for IHC/IF staining were as follows: rabbit anti-Caspase-3, rabbit anti-GAS6, rabbit anti-Axl, rabbit anti-MMP13, rabbit anti-Aggrecan, mouse anti-F4/80, mouse anti-iNOS, rabbit anti-CD206, species-matched horseradish peroxidase-conjugated secondary antibodies, and species-matched Alexa-488- or 594-labeled secondary antibody.

## Cartilage and synovium structure grading

Histology sections of the knee joints were graded based on the Osteoarthritis Research Society International (OARSI) scoring system developed by *Glasson et al., 2010* by two observers blinded to the experimental conditions. Generally, sections were assigned a grade of 0–6: 0, normal cartilage; 0.5, slight loss of Safranin O staining without structural changes; 1, small fibrillations without loss of cartilage; 2, vertical clefts down to the layer below the superficial layer; 3–6, vertical clefts or erosion to the calcified cartilage affecting <25% (grade 3), 25–50% (grade 4), 50–75% (grade 5), and >75% (grade 6) of the articular surface. Synovitis severity was estimated based on the synovial lining cell layer enlargement, resident cell density, and inflammatory infiltration. A nine-point scale was used, where low scores indicated moderate synovitis and high scores represented severe synovitis (*Krenn et al., 2006*).

## TUNEL

The TUNEL assay was performed according to the manufacturer's instructions (TUNEL Apoptosis Detection Kit [Alexa Fluor 488], Yeasen, #40307ES20) to detect cell death in the synovial membrane. The assay used the green channel at 488 nm. DAPI was applied as a nuclear counterstain in the blue channel at 461 nm. Images were taken with an *Olympus BX43* fluorescent microscope and *Olympus DP73* digital camera at ×400 magnification with cellSens software (Olympus). Exposure settings were adjusted to minimize oversaturation.

## Apoptosis induction and thymocyte IF staining

Murine thymocytes were isolated from C57BL/6 mice and then stimulated with 25 µmol/l dexamethasones for 3 hr to induce apoptosis, followed by washing twice and resuspension with phagocyte culture medium to a concentration of $1 \times 10^7$ cells/ml. Then, thymocytes were labeled with CFSE dye following the manufacturer's instructions.

## Efferocytosis assay

An efferocytosis assay was performed as previously described (*Heo et al., 2014*). Briefly, RAW264.7 or BMDM cells were plated in a 6-well plate ($1 \times 10^6$ cells/well) with Dulbecco's modified Eagle medium containing 10% fetal bovine serum and cultured overnight. RAW264.7 cell line was purchased from Pricell Life Science & Technology. The cells were authenticated by STR profiling and were not contaminated by mycoplasma. The cells were then treated with LPS, rmGAS6, or R428 for 24 or 48 hr. CFSE-labeled apoptotic thymocytes were then added at a ratio of 10:1 and incubated for an additional 120 min. The cells were extensively washed three times with PBS to remove unengulfed thymocytes. The ability of macrophages to engulf apoptotic thymocytes was quantified by flow cytometry or visualized using IF microscopy. The efferocytotic index was calculated using the following formula: (number of macrophages containing apoptotic bodies)/(total macrophages) × 100% and then normalized using the control group as 100%.

## Real-time polymerase chain reaction

Total RNA was isolated from RAW264.7 cells and ground cartilage from human tibial plateaus using TRIzol reagent. For mRNA quantification, 1 mg of total RNA was purified with gDNA remover and reverse transcribed using 5× HiScript II qRT SuperMix II. Each PCR reaction consisted of 10 µl of 2×

ChamQ SYBR qPCR Master Mix, 10 µM of forward and reverse primers, and 500 ng of cDNA. For miRNA quantification, 1 mg of total RNA was purified with gDNA wiper mix and then reverse transcribed using Hiscript II Enzyme Mix, 10× RT Mix, and specific stem-loop primers. Template DNA was mixed with 2× miRNA Universal SYBR qPCR Master Mix, specific primers, and mQ primer R. All reactions were run in triplicate. Mouse primer sequences are listed in the Key Resources Table.

### Enzyme-linked immunosorbent assay

Human synovial fluid samples were collected as described above. All samples were spun down at 4500 × *g* for 15 min. Human GAS6 Quantikine Kit (R&D Systems) was used to measure the concentration of GAS6 in the synovial fluid.

### Statistical analyses

Data were represented as the mean ± standard deviation. An unpaired Student's *t*-test was performed for experiments comparing two groups of data. A one-way analysis of variance was performed for data involving multiple groups, followed by Tukey's post hoc test. p values of <0.05 were considered statistically significant.

## Acknowledgements

This work was supported by grants from the National Natural Science Foundation of China (grant numbers: 81902229 and 81871745) and Natural Science Foundation of Guangdong Province (2020A1515011062).

## Additional information

### Funding

| Funder | Grant reference number | Author |
| --- | --- | --- |
| National Natural Science Foundation of China | 81902229 | Haiyan Zhang |
| National Natural Science Foundation of China | 81871745 | Anling Liu |
| Natural Science Foundation of Guangdong Province | 2020A1515011062 | Haiyan Zhang |

The funders had no role in study design, data collection, and interpretation, or the decision to submit the work for publication.

### Author contributions

Zihao Yao, Data curation, Investigation, Writing - original draft; Weizhong Qi, Data curation, Software, Formal analysis; Hongbo Zhang, Conceptualization, Software, Methodology; Zhicheng Zhang, Validation, Investigation; Liangliang Liu, Resources, Validation, Investigation; Yan Shao, Software, Methodology; Hua Zeng, Resources, Software; Jianbin Yin, Supervision, Methodology; Haoyan Pan, Supervision, Visualization, Methodology; Xiongtian Guo, Software, Validation; Anling Liu, Supervision, Funding acquisition; Daozhang Cai, Conceptualization, Writing - review and editing; Xiaochun Bai, Software, Validation, Methodology; Haiyan Zhang, Conceptualization, Formal analysis, Funding acquisition

### Author ORCIDs

Xiaochun Bai (ID) http://orcid.org/0000-0001-9631-4781
Haiyan Zhang (ID) http://orcid.org/0000-0002-9361-3134

### Ethics

Informed consent was obtained from all recruited patients and was identified by the ethics committee of the Third Affiliated Hospital of Southern Medical University.

This study was performed in strict accordance with the recommendations in Chinese Laboratory animal-Guideline for ethical review of animal welfare (GB/T 35892-2018). The protocol was approved by the Southern Medical University Animal Care and Use Review Board (Permit Number: 2021-Ethical review-053). All surgery was performed under sodium pentobarbital anesthesia, and every effort was made to minimize suffering.

## Decision letter and Author response

Decision letter https://doi.org/10.7554/eLife.83069.sa1
Author response https://doi.org/10.7554/eLife.83069.sa2

---

## Additional files

### Supplementary files

• MDAR checklist

### Data availability

Sequencing data have been deposited in GEO under accession code GSE53986. Source Data has been uploaded in Dryad, which was named after 'Down-regulated GAS6 impairs synovial macrophage efferocytosis and promotes obesity-associated osteoarthritis' (https://doi.org/10.5061/dryad.d2547d86d).

The following dataset was generated:

| Author(s) | Year | Dataset title | Dataset URL | Database and Identifier |
|---|---|---|---|---|
| Zihao Y | 2023 | Down-regulated GAS6 impairs synovial macrophage efferocytosis and promotes obesity-associated osteoarthritis | https://dx.doi.org/10.5061/dryad.d2547d86d | Dryad Digital Repository, 10.5061/dryad.d2547d86d |

The following previously published dataset was used:

| Author(s) | Year | Dataset title | Dataset URL | Database and Identifier |
|---|---|---|---|---|
| Noubade R | 2014 | NRROS negatively regulates ROS in phagocytes during host defense and autoimmunity | https://www.ncbi.nlm.nih.gov/geo/query/acc.cgi?acc=GSE53986 | NCBI Gene Expression Omnibus, GSE53986 |

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
