## [Editor Report]

In this study, the authors demonstrated that patients with obese-OA and mice with ApoE deficiency showed phenotypes of synovitis and enhanced macrophage infiltration in synovial tissues. GAS6 secretion is decreased during M1 macrophage polarization during obese-OA, leading to impaired macrophage efferocytosis in synovial apoptotic cells. Intra-articular injection of GAS6 restored the phagocytic capacity of macrophages, decreased synovial cell apoptosis, and prevented OA progression in obese-OA mice.

---

## [Decision Letter]

**Decision letter after peer review:**

Thank you for submitting your article "Down-regulated GAS6 impairs synovial macrophage efferocytosis and promotes obesity-associated osteoarthritis" for consideration by *eLife*. Your article has been reviewed by 3 peer reviewers, one of whom is a member of our Board of Reviewing Editors, and the evaluation has been overseen by Mone Zaidi as the Senior Editor. The following individual involved in the review of your submission has agreed to reveal their identity: Chao Xie (Reviewer #2).

In this study, the authors demonstrated that patients with obesity-associated osteoarthritis and mice with the ApoE gene deficiency showed phenotypes of synovitis and enhanced macrophage infiltration in synovial tissues.

GAS6 is a glycoprotein and its secretion is decreased during M1 macrophage polarization during obesity-associated osteoarthritis, leading to impaired macrophage efferocytosis in synovial apoptotic cells.

Intra-articular injection of GAS6 restored the phagocytic capacity of macrophages, decreased synovial cell apoptosis, and prevented osteoarthritis progression in mice with obesity-associated osteoarthritis.

Essential Revisions:

1) The authors need to carefully explain the rationale behind their design of the experimental groups.

2) The authors also need to analyze changes in macrophage efferocytosis maker F4/80 in OA synovitis.

3) If the administration of GAS6 can also reverse OA-like feasure in human cartilage explants or human articular chondrocytes.

*Reviewer #1 (Recommendations for the authors):*

The authors need to address the comments below.

1) If the administration of GAS6 can also reverse OA-like feasure in human cartilage explants or human articular chondrocytes.

2) To determine that GAS6/Axl signaling is indeed involved in OA development, the authors may need to provide evidence about the role of Axl in OA development.

*Reviewer #2 (Recommendations for the authors):*

The manuscript entitled "Down-regulated GAS6 impairs synovial macrophage efferocytosis and promotes obesity-associated osteoarthritis" by Dr. Yao et al. observed that M1 macrophage infiltration significantly increased in the synovial tissue of obesity-associated OA. Overall, this is a well-designed, highly clinical demand and potentially translational study. Based on the patient's sample, the data indicated Synovial tissues are highly hyperplastic in obese OA patients and infiltrated with more polarized M1 macrophages than in non-obese OA patients.

Further authors proved that obesity promotes synovial M1 macrophage accumulation and GAS6 was inhibited in synovitis during OA development in mice models. The sample size, data collection, and quality of the IHC and immunofluorescent histological sections are outstanding. The results were well presented with appropriate interpretation.

*Reviewer #3 (Recommendations for the authors):*

The main strengths of the paper are the discovery of the underlying mechanism of obesity-associated osteoarthritis. However, some claims and conclusions were not well supported by their data. There are some issues that need to be carefully clarified and studied:

1. The design of experimental groups was defective, all C57BL/6 mice were fed a standard diet, while ApoE-/- mice were fed a high-fat diet. Both C57BL/6 and ApoE-/- mice should be fed the standard and high-fat diet respectively.

2. The source of cartilage and synovial tissue of obese OA patients described in Figure 1 should be added to the Materials and methods.

3. The authors deemed that the expression of GAS6 in chondrocytes was decreased in the obese ApoE-/- OA mice, however, in Figure 3C, its expression seemed to be increased.

4. The authors declared that GAS6 expression was inhibited in synovial macrophages, whether GAS6 is mainly expressed in synovial macrophages? Why was the expression level decreased in chondrocytes?

5. In Figure 1B, the F4/80 labeled macrophages were increased in synovial tissues of obese OA patients, however, in Figure 3A-D, the expression of F4/80 in synovial tissues of obese OA seemed to be decreased.

6. In Figure 4 H-J, the authors should add a group that is stimulated with GAS6 and R428 in RAW264.7 cells to prove that inhibition of the Axl receptor by R428 diminished the enhanced phagocytotic activity.

7. The authors declared that blocking M1 macrophage polarization could be a potential therapeutic strategy for obesity-associated OA, however, the authors did not investigate the effect of blocking M1 macrophage polarization.

8. The authors declared that enhanced M1-polarized macrophages in obese synovium decreased GAS6 secretion, the evidence for this conclusion was weak, and the decreased GAS6 secretion maybe not be due to the polarization of macrophages.

9. To study macrophage efferocytosis, the markers F4/80 should be used, but not the M1 macrophage marker iNOS. Besides, the primary macrophage such as BMDM is better than 264.7 cells.

10. The authors thought that accumulated apoptotic cells lead to the release of inflammatory factors that induced chondrocyte homeostasis dysfunction in obese OA patients, however, additional evidence is needed to support this conclusion.

---

## [Author Response]

Essential Revisions:1) The authors need to carefully explain the rationale behind their design of the experimental groups.

We greatly appreciate the careful review and helpful suggestions for improving manuscript. The experimental group was divided into four groups according to disease model and genotype: C57BL/6 control group, C57BL/6 OA group, ApoE^-/-^ control group and ApoE^-/-^ OA group.

From a critical perspective, the experimental grouping should be more rigorous in the manuscript. According to previous studies which revealed that feeding ApoE^-/-^ mice with HFD accelerates the increase in LDL cholesterol levels and causes more body weight gain, compared with C57BL/6 mice fed with an HFD (references were provided in Answer 1# for Reviewer #2). We regard ApoE^-/-^ mice fed with an HFD as the total variable associated with obesity, while C57BL/6 mice fed with a standard-chow diet as control. During revising the article, we reared mice and divided them into four groups, as Reviewer #3 suggested. Body weight gain was re-analyzed and provided in revised Table 4. Our statistical data also revealed a significant increase in body weight by feeding ApoE^-/-^ mice with an HFD (19.81±1.33g) for 8 weeks compared with feeding C57BL/6 mice with an HFD (16.89±0.75g), which partly verified the effectiveness of inducing obesity by feeding ApoE^-/-^ mice with an HFD for a short-term period. Accordingly, we selected feeding ApoE^-/-^ mice with an HFD as the experimental group while feeding C57BL/6 mice with a standard-chow diet as control. Moreover, further studies were designed to explore the mechanism of obesity in the progression of OA.

2) The authors also need to analyze changes in macrophage efferocytosis maker F4/80 in OA synovitis.

Thanks for this valuable comment. Previous studies showed that MER is closely related to the maintenance of macrophages efferocytosis. Dransfield et al. identified Mer as a receptor uniquely capable of tethering ACs to the macrophage surface and driving their subsequent internalization^1^. Moreover, Felton and colleagues further revealed that knocking out Mer impaired phagocytosis of mouse eosinophils by macrophages, contributing to the persistence of airway inflammation^2^. Cai et al. found that higher macrophage MerTK improved efferocytosis and increased the ratio of pro-resolving versus proinflammatory lipid mediators, which attenuated plaque inflammation, fibrous cap thinning, and thrombosis in atherosclerosis^3^. These results suggest that MER is essential for the normal functioning of macrophage efferocytosis. In this manuscript, we found a decreased expression of MER in OA hyperplastic synovial tissue, especially in obesity-associated OA. These results indicated that the efferocytosis of macrophages is impaired in obesity-associated OA.

We have now added this part of the results in Revised Results 4, which reads “MER were widely used to label macrophages undergoing efferocytosis. Immunofluorescence staining analysis indicated the expression of MER in OA synovial macrophages decreased significantly, especially in obese ApoE^-/-^ OA and obese OA patients.” We have now included these results in the revised Figure 4—figure supplement 1.

3) If the administration of GAS6 can also reverse OA-like feasure in human cartilage explants or human articular chondrocytes.

Thanks for this insightful comment. We have now used both human cartilage explants and primary articular chondrocytes obtained from patients undergoing total knee arthroplasty to confirm the effect of GAS6 on chondrocytes. Patient consent and the approval of the Ethics Committee of the Third Affiliated Hospital of Southern Medical University (Guangzhou, China) were obtained before the human tissue samples were harvested. No apparent proteoglycan loss or increase was found in rhGAS6 treated cartilage explant. Moreover, Western blotting showed no significant differences in the expression of COL2, MMP13, p16, and p21 in primary chondrocytes after stimulation with rhGAS6. These data are shown in revised Figure 5—figure supplement 2 and Figure 5—figure supplement 3

Reviewer #1 (Recommendations for the authors):The authors need to address the comments below.1) If the administration of GAS6 can also reverse OA-like feasure in human cartilage explants or human articular chondrocytes.

Thank you for this valuable comment, and thank the Reviewer very much for the opportunity to improve the manuscript. Please refer to Response to Essential Revisions comments Question 3.

2) To determine that GAS6/Axl signaling is indeed involved in OA development, the authors may need to provide evidence about the role of Axl in OA development.

Thank you for this insightful comment. As shown in our manuscript, F4/80-Axl double staining revealed no significant difference in the number of Axl-positive cells in the synovium of obese OA patients, obese ApoE^-/-^ OA mice, and control subjects (previous Figure 3—figure supplement 3). Articular administration of R428 (Axl inhibitor) in C57BL/6 OA mice promoted synovial hyperplasia and cartilage destruction and enhanced synovitis and OARSI scores (previous Figure 5A and B). Moreover, in vitro experiments showed that the phagocytotic activity of RAW264.7 cells was decreased after administration of R428 (previous Figure 4H).

As the reviewer points out, more evidence is required to support the role of Axl in OA development. To demonstrate the direct role of Axl in macrophage, R428 was administrated to Raw264.7 cells to inhibit Axl. The results showed an increased expression of iNOS and CD86, together with up-regulated IL-1、IL-6, and TNF-α. These data are shown in revised Figure 3—figure supplement 2C and Figure 4—figure supplement 3.

Reviewer #3 (Recommendations for the authors):The main strengths of the paper are the discovery of the underlying mechanism of obesity-associated osteoarthritis. However, some claims and conclusions were not well supported by their data. There are some issues that need to be carefully clarified and studied:1. The design of experimental groups was defective, all C57BL/6 mice were fed a standard diet, while ApoE-/- mice were fed a high-fat diet. Both C57BL/6 and ApoE-/- mice should be fed the standard and high-fat diet respectively.

Thank you for this valuable comment, and thank the Reviewer very much for the opportunity to improve the manuscript. From a critical perspective, the experimental grouping should be more rigorous in the manuscript. According to previous studies which revealed that feeding ApoE^-/-^ mice with HFD accelerates the increase in LDL cholesterol levels and causes more body weight gain, compared with C57BL/6 mice fed with an HFD (references were provided in Answer 1# for Reviewer #2). We regard ApoE^-/-^ mice fed with an HFD as the total variable associated with obesity, while C57BL/6 mice fed with a standard-chow diet as control. During revising the article, we reared mice and divided them into four groups, as you suggested. Body weight gain was re-analyzed and provided in revised Table 4. Our statistical data also revealed a significant increase in body weight by feeding ApoE^-/-^ mice with HFD (19.81±1.33g) for 8 weeks compared with feeding C57BL/6 mice with an HFD (16.89±0.75g), which partly verified the effectiveness of inducing obesity by feeding ApoE^-/-^ mice with an HFD for a short-term period. Accordingly, we selected feeding ApoE^-/-^ mice with an HFD as the experimental group while feeding C57BL/6 mice with a standard-chow diet as control. Moreover, further studies were designed to explore the mechanism of obesity in the progression of OA.

2. The source of cartilage and synovial tissue of obese OA patients described in Figure 1 should be added to the Materials and methods.

Thank you for your suggestions. We have added the source of cartilage and synovial tissue from obese OA patients in the first section of Materials and methods section. It reads “Synovial tissue and synovial fluid samples of normal individuals (n = 6, age 34 ± 8.15 years, three males, three females) or obese individuals (n = 6, age 35 ± 7.36 years, four males, two females) were obtained from patients who received arthroscopic treatment for acute anterior cruciate ligament rupture or meniscus injury Other joint diseases were excluded from the study. OA synovial tissues, and synovial fluid samples were obtained from obese patients (n = 6, age 64 ± 5.75 years, two males, four females) or patients without obesity (n = 6, age 65 ± 4.26 years, three males, three females) who underwent total knee arthroplasty. OA cartilage tissues were obtained from patients who underwent total knee arthroplasty, the tibial plateau cartilage was carefully separated and cut into 1mm^3^, cultured in vitro.”

3. The authors deemed that the expression of GAS6 in chondrocytes was decreased in the obese ApoE-/- OA mice, however, in Figure 3C, its expression seemed to be increased.

Thank you for your comments. We have replaced it with more representative images.

4. The authors declared that GAS6 expression was inhibited in synovial macrophages, whether GAS6 is mainly expressed in synovial macrophages? Why was the expression level decreased in chondrocytes?

We greatly appreciate your careful review and helpful suggestions. Our previous results showed that both macrophages and chondrocytes expressed GAS6 protein. However, we found a decrease in chondrocyte protein expression but did not investigate its functional impact. As suggested by Reviewer 1, rhGAS6 were used to stimulate OA cartilage explants or primary chondrocytes, we found that rhGas6 may not reverse the dysregulation of anabolic and catabolic homeostasis in OA articular cartilage explants, these data are shown in revised Figure 5—figure supplement 2 and Figure 5—figure supplement 3 (referring to answer for Essential Revisions 3). Since administration of GAS6 has no obvious effect on chondrocyte metabolism, we delete this part of figures. Nevertheless, the underlying mechanism of the decreased expression of GAS6 deserves further investigation.

5. In Figure 1B, the F4/80 labeled macrophages were increased in synovial tissues of obese OA patients, however, in Figure 3A-D, the expression of F4/80 in synovial tissues of obese OA seemed to be decreased.

We greatly appreciate your careful review and helpful suggestions. We are sorry for the ambiguity caused by the quality of images and we have chosen more representative images in revised Figure 3A-D.

6. In Figure 4 H-J, the authors should add a group that is stimulated with GAS6 and R428 in RAW264.7 cells to prove that inhibition of the Axl receptor by R428 diminished the enhanced phagocytotic activity.

Thank you for your suggestions. We added the group treated with both GAS6 and R428. The phagocytotic activity was indeed diminished after stimulated with rmGAS6 and R428 compared administration with rmGAS6 alone. We have now included these results in revised Figure 4H-K.

7. The authors declared that blocking M1 macrophage polarization could be a potential therapeutic strategy for obesity-associated OA, however, the authors did not investigate the effect of blocking M1 macrophage polarization.

We greatly appreciate your careful review and helpful suggestions. We are sorry for the ambiguity caused by the expression. Our previous findings showed that blocking M1 polarization of synovial macrophages attenuated OA progression. In this manuscript, we revealed that M1-polarized macrophage infiltration in OA synovial tissue of obese patients is significantly increased. Nevertheless, evidence supporting the effect of blocking M1 polarization could attenuate OA development is insufficient. We have deleted such ambiguous expressions and corrected the expression in the manuscript, which reads “targeting macrophage associated efferocytosis or intra-articular injection of GAS6 is a potential therapeutic strategy for obesity-associated OA.”

8. The authors declared that enhanced M1-polarized macrophages in obese synovium decreased GAS6 secretion, the evidence for this conclusion was weak, and the decreased GAS6 secretion maybe not be due to the polarization of macrophages.

We greatly appreciate your careful review and helpful suggestions. In order to explore the effect of M1 macrophage polarization on GAS6 secretion, BMDMs were treated with LPS or IL-4 for 36 hours. As is shown in revised Figure 3—figure supplement 2B, the expression of iNOS and CD86 were increased while GAS6 decreased significantly after treated BMDMs with LPS for 36 hours. These data indicated that the decreased GAS6 secretion was mainly due to the M1 polarization of macrophages.

9. To study macrophage efferocytosis, the markers F4/80 should be used, but not the M1 macrophage marker iNOS. Besides, the primary macrophage such as BMDM is better than 264.7 cells.

Thank you for your comment. In Figure 4E-J, marker F4/80 (red) and CFSE (green) were used to label macrophages and apoptotic cells respectively. Moreover, as you proposed, we performed an experiment with BMDMs to illustrate the effect of rmGAS6 in regulating efferocytosis of macrophages, which were showed in Revised Figure 4—figure supplement 2.

10. The authors thought that accumulated apoptotic cells lead to the release of inflammatory factors that induced chondrocyte homeostasis dysfunction in obese OA patients, however, additional evidence is needed to support this conclusion.

Thank you for your comment. To further explore the effect of inflammatory factors released by the stimulation of apoptotic cells (ACs) on chondrocyte homeostasis dysfunction, ACs were co-cultured with BMDMs, and the culture supernatant was collected after stimulation for 24 hours. The expression of MMP13 was increased after adding co-culture supernatant stimulation, together with senescence hallmarks such as p16 and p21, while COL2 were down-regulated after stimulation. These data partly indicated that macrophages stimulated by apoptotic cells secreted chemokines leading chondrocyte homeostasis dysfunction. However, the effect of supernatant promoting chondrocyte homeostasis dysfunction was partially alleviated when adding ACs with rmGAS6 to BMDMs for 24 hours, which were diminished by adding R428. We adding these contexts in Results 5, which reads “To further explore the effect of inflammatory factors released by the stimulation of apoptotic cells (ACs) on chondrocyte homeostasis dysfunction, ACs were co-cultured with BMDMs and the culture supernatant was collected after stimulation for 24 hours. The expression of MMP13 were increased after adding co-culture supernatant stimulation, together with senescence hallmarks such as p16 and p21, while COL2 were down-regulated after stimulation. However, the effect of supernatant promoting chondrocyte homeostasis dysfunction was partially alleviated when adding ACs with rmGAS6 to BMDMs, which were diminished by adding R428.” These data were showed in revised Figure 5—figure supplement 1.